# Characterization of Ancient Burial Pottery of Ban Muang Bua Archaeological Site (Northeastern Thailand) Using X-ray Spectroscopies

Chatdanai Boonruang [1,2], Krit Won-in [3,*], Nontarat Nimsuwan [3], Phakkhananan Pakawanit [4], Udomrat Tippawan [5], Chome Thongleurm [6] and Pisutti Dararutana [7,†]

1   Department of Physics and Materials Science, Faculty of Science, Chiang Mai University, Chiang Mai 50200, Thailand; chatdanai.b@cmu.ac.th
2   Center of Excellence in Materials Science and Technology, Chiang Mai University, Chiang Mai 50200, Thailand
3   Department of Earth Sciences, Faculty of Science, Kasetsart University, Bangkok 10900, Thailand; ryukazeii@gmail.com
4   Synchrotron Light Research Institute, Nakhon Ratchasima 30000, Thailand; phakkhananan@slri.or.th
5   Plasma and Beam Physics Research Facility, Department of Physics and Materials Science, Faculty of Science, Chiang Mai University, Chiang Mai 50200, Thailand; udomrat.t@cmu.ac.th
6   Science and Technology Research Institute, Chiang Mai University, Chiang Mai 50200, Thailand; chome.t@cmu.ac.th
7   Independent Researcher, Lopburi 15000, Thailand; pisutti@hotmail.com
*   Correspondence: kritwonin@gmail.com
†   Retired Army Officer at the Royal Thai Army.

**Abstract:** Ancient potteries found at Ban Muang Bua, located in northeastern Thailand, associate with Thung Kula Ronghai culture. Most of them are products used in daily life and grave goods. The potsherds were examined using techniques based on X-ray spectroscopy. Elemental composition and morphology were analyzed using proton-induced X-ray emission spectroscopy (PIXE) and scanning electron microscopy coupled with energy-dispersive X-ray spectroscopy (SEM-EDS). Three-dimensional analysis was performed using X-ray tomographic microscopy based on synchrotron radiation (SR XTM). Irregular plate-like particles of composites with a wide range of size distribution were found in the potsherds. The major (O, Si, and Al), minor (C, Fe, Ca, and K), and trace elements (P, S, Ti, Na, Mg, and Zn) which were observed can provide the information about raw materials and production of pottery. The 3D tomographic images show the internal feature of these samples. The combination of SEM-EDS, PIXE, and SR XTM is a powerful tool for archaeological research especially in terms of composition and internal structure. The results imply that the raw materials of pottery were sandy soil derived from marine sands, clays, and salt deposits that were mostly iron-rich-kaolin clay. The production was carried out with low firing temperatures (~600–900 °C) in open-air kilns.

**Keywords:** ancient burial pottery; Ban Muang Bua; SEM-EDS; PIXE; SR XTM

## 1. Introduction

Research in cultural heritage objects such as ceramics, mortars, glasses, etc., is multi-disciplinary work that incorporates researchers from various fields of study. The identification and classification in historical and geographical contexts of these artistic objects are important [1,2]. Analytical techniques for these objects should possess non-destructive operation, high sensitivity, spatial resolution, multi-elemental analysis, and versatility for providing reliable information (in micro to nanoscales) about the complex structure of cultural heritage objects. Quantitative and/or semi-quantitative analyses for major, minor, and trace elements of archaeological materials are required to obtain information about manufacturing technology, raw materials, and origin of these objects as well as their restoration and conservation [3–5]. In previous research, many X-ray based techniques have

been successfully used for analyses of the archaeological objects including X-ray fluorescence spectroscopy (XRF), X-ray diffraction (XRD), proton-induced X-ray emission (PIXE), X-ray photoelectron spectroscopy (XPS), X-ray absorption spectroscopy (XAS), and X-ray tomographic microscopy (XTM) [3,6–11].

It is reported that the ceramics which are inorganic and non-metallic solids were produced by firing and cooling processes. It is well-known that clay is the raw material for pottery production, which contains fine particles and minerals that are weather-resistant materials of igneous rocks.

There are many works about characterization of ancient ceramics found in Thailand including brick, jar, and pottery at different archaeological sites such as Wiang Kaen, Mae Man Noi, U-Thong, Wiang Kalong, Ban Chiang, and Ban Muang Bua [12–20].

Jar burial cultures associate mainly with the tradition of burying which includes not only the human body (primary burial) but also human bones (secondary burial). These burials have been found in Asia, such as in Japan, Taiwan, China, Korea, Laos, Malaysia, Indonesia, and Philippines which are dated to at least 1000 B.C. It has been reported that buried grave jars excavated from Mun and Chi valleys in northeastern Thailand are dated from the late prehistoric period (2000–500 B.C.) to 11th century A.D. [21,22].

Ban Muang Bua archaeological site, Muang Bua sub-district, Kasetwisai district, Roi-Et province, northeastern Thailand is located at Geographic Latitude 15.607838° N and Geographic Longitude 103.592923° E on Thung Kula Rong Hai area of Korat basin (1500 B.C.–500 A.D.). The area covers some parts of the five provinces of Roi-Et, Mahasarakham, Surin, Srisaket, and Yasothorn, as shown in Figure 1a. It was chosen to be the main site for research in burial pottery, grave goods, and funeral ceremonies in the late prehistorical age due to possession of archaeological evidence and becoming a restricted area to the intrusion of agricultural activities. Grave goods and products used in daily life have been discovered at this site. These local unique products are called Roi Et earthenware. The burial jars have been used in both primary and secondary burial ceremonies. Most of them are low-temperature fired earthenware with thicknesses of ~1–3 cm. They are also decorated with cord-marked, red slipped, color painted, and molding paste patterns [23] as shown in Figure 1b and Table 1. Moreover, archaeological evidence reveals that the Ban Muang Bua archaeological site is contemporary with Ban Chiang and others. Carbon-14 dating (in 2010) confirmed that the pottery samples excavated in between 2002–2003 are dated to 3890 ± 260 to 2620 ± 240 years for primary burial and 2440 ± 210 to 1490 ± 210 years for secondary burial [23–25]. Thermoluminescence dating of the pottery samples (of secondary burial) excavated in 2003 is of between 2045 ± 50 and 1650 ± 40 years (on 30 July 2020) [26].

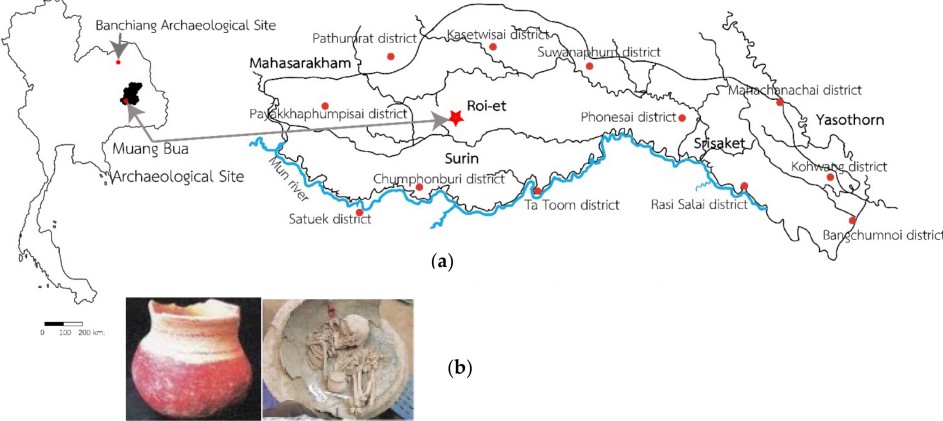

(a)

(b)

**Figure 1.** (**a**) Thung Kula Rong Hai area, and (**b**) ancient burial pottery at Ban Muang Bua archaeological site. Adapted with permission from ref. [24] (2010) (Fine Arts Department: Ubol Ratchathani, Thailand) and Nimsuwan, N.; Dutchaneephet, J.; Pakawanit, P.; Dararutana, P.; Won-in, K. (2019) (Malays. J. Microsc.) [20].

**Table 1.** Description of MB051, MB052, MB054, MB055, MB056, MB058, MB059, and MB060.

| Sample | Description | | Test Pit | Estimated Date (Years) [24] |
| | Typology | Thickness (cm) | | |
| --- | --- | --- | --- | --- |
| MB051 | | 1.1 | MB'2003 PK.PIT 3 S6W2 Burial jar No 1/63 | |
| MB052 | | 1.0 | MB'2003 PK.PIT 3 S3W2-3 Burial jar No. 1/112 | |
| MB054 | | 0.8 | MB'2003 PK.PIT 5 N1-2 W1-2 Burial jar No. 187 | |
| MB055 | :Body sherd :Plain with red-slipped body sherd | 0.6 | MB'2003 PK.PIT 5 N2W3 Burial jar No. 188 | 2440 ± 210 to 1490 ± 210 |
| MB056 | | 0.6 | MB'2003 PK.PIT 2 N2E4 Burial jar No. 132 | |
| MB058 | | 0.8 | MB'2003 PK.PIT 4 S2E3 Burial jar No. 123 | |
| MB059 | | 1.2 | MB'2003 PK.PIT 4 S2E3 Burial jar No. 119 | |
| MB060 | | 1.2 | MB'2003 PK.PIT 4 S2E5-6 Burial jar No. 117 | |

In this work, a set of samples were selected from the ancient potsherds of grave goods (of secondary burial) excavated from the Ban Muang Bua archaeological site in Roi Et province in 2002–2003. The morphology and elemental composition of samples were characterized by SEM-EDS, PIXE, and SR XTM with the aim to compare the composition to those from our previous work (excavated in 2002–2003 with similar typology) that can lead to the information about raw materials and production technology of the pottery.

## 2. Materials and Methods

### 2.1. Materials

Eight samples were found at the mound (approximately 4 m in height) behind temple of Wat Phathum Khongkha which was a central monastery of Ban Muang Bua [24]. The samples were each selected from 8 different sample pits (Figure 2) and labeled as MB051, MB052, MB054, MB055, MB056, MB058, MB059, and MB060 (Figure 3). They possessed the same typology but different in thickness. The details are shown in Table 1. Each sample consisted of three layers of the outer layer, core, and inner layer. The outer and inner layers were produced for surface protection and wear resistance [20].

### 2.2. Methods

The contents of elements in samples were analyzed and classified as major, minor, and trace elements using SEM-EDS (with the contents of $\geq 0.10$ wt%), while PIXE confirmed and focused on trace elements in ppm.

A QUANTA 450 (FEI Company s.r.o., Brno, Czech Republic) scanning electron microscope (SEM) coupled with X-Max EDS System (Oxford Instruments, Oxford, UK) energy-dispersive X-ray spectrometer (EDS) at Science Equipment Center of Faculty of Science (Kasetsart University, Bangkok, Thailand) was carried out to characterize the microstructure

and elemental composition of samples. The SEM was operated at 15 kV with magnification ranging from 25x to 500x. Semi-quantitative analysis of EDS spectra was carried out using INCA software (Oxford Instruments, Oxford, UK). X-ray spectra were recorded from three different positions for each sample: the outer layer; core; and inner layer as shown in Figure 3.

Proton-induced X-ray emission spectroscopy (PIXE) (operated at Plasma and Beam Physics Research Facility of Chiang Mai University, Chiang Mai, Thailand) was applied to identify and analyze contents of elements, especially the trace ones. The analysis was based on 2-MeV-proton beam produced by 1.7 MV tandem Tandetron accelerator (Model 4117, High Voltage Engineering Europe Corp., Utrecht, the Netherlands). The proton beam was collimated with a diaphragm of 1 mm in diameter. The beam current on sample was controlled to be of 10 nA. The sample was probed at the same position of SEM-EDS using a signal detector of Si(Li) with 6.2 mm in diameter and 3 mm thick (Model SL30170, Canberra Industries Inc., Connecticut, USA). Silicon with known concentration was used as a standard reference for PIXE analysis as conducted in previous research [27–33]. Quantitative analysis of chemical composition (for $Z \geq 13$) was performed using GUPIXWIN code (GUPIX software for Windows version, Department of Physics, University of Guelph, Ontario, Canada) [34–36]. The quantitative calibration includes normalization at (100%-x) of the total amount of oxides, where the value of x is a sum of $Na_2O$ and MgO determined by EDS. The subtraction of x was due to the accuracy of detection of Na and Mg by PIXE was not good when compared to other elements with $Z \geq 13$.

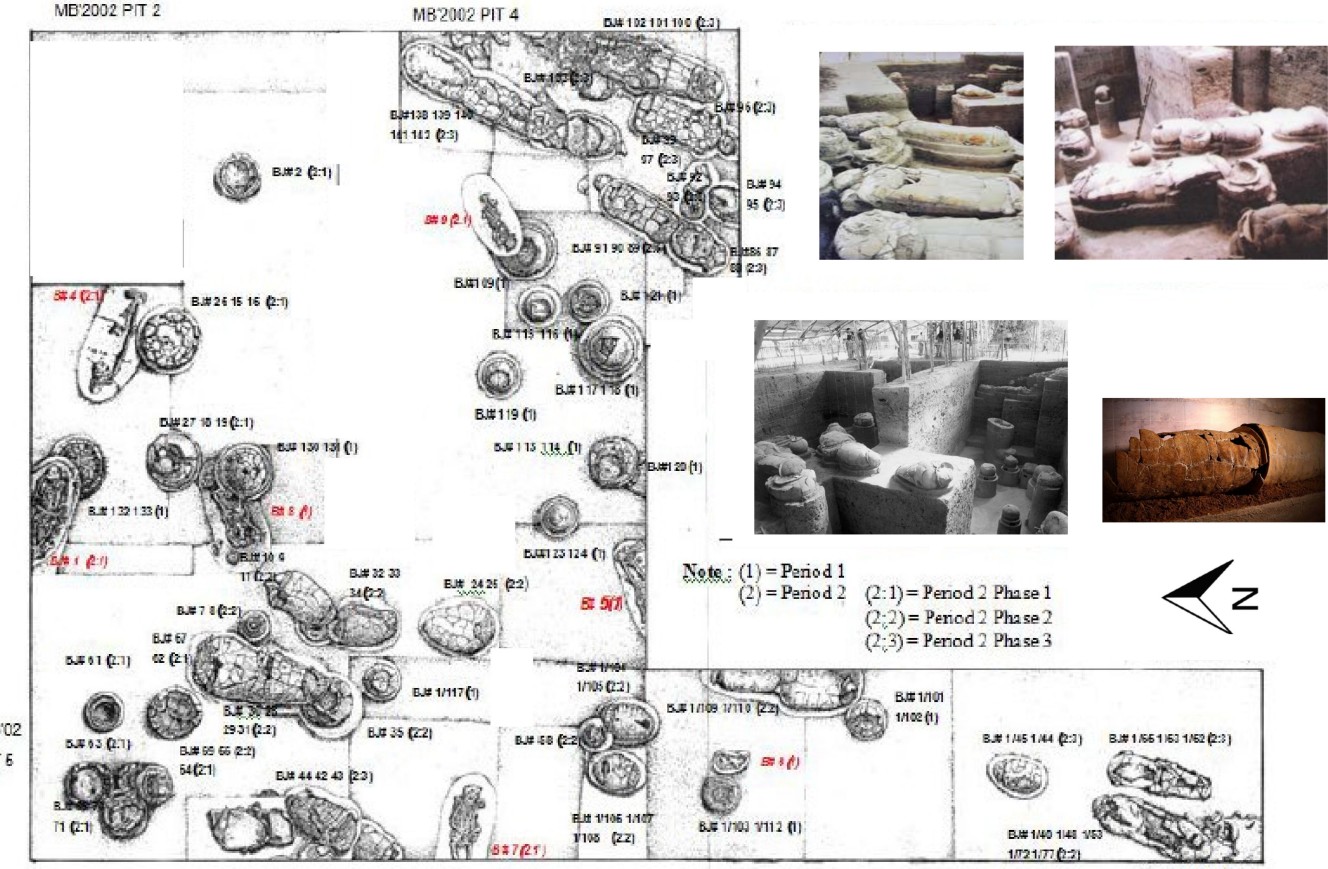

**Figure 2.** Plan of jar burial sample pits at Ban Muang Bua. Adapted with permission from ref. [24] (2010) (Fine Arts Department: Ubol Ratchathani, Thailand).

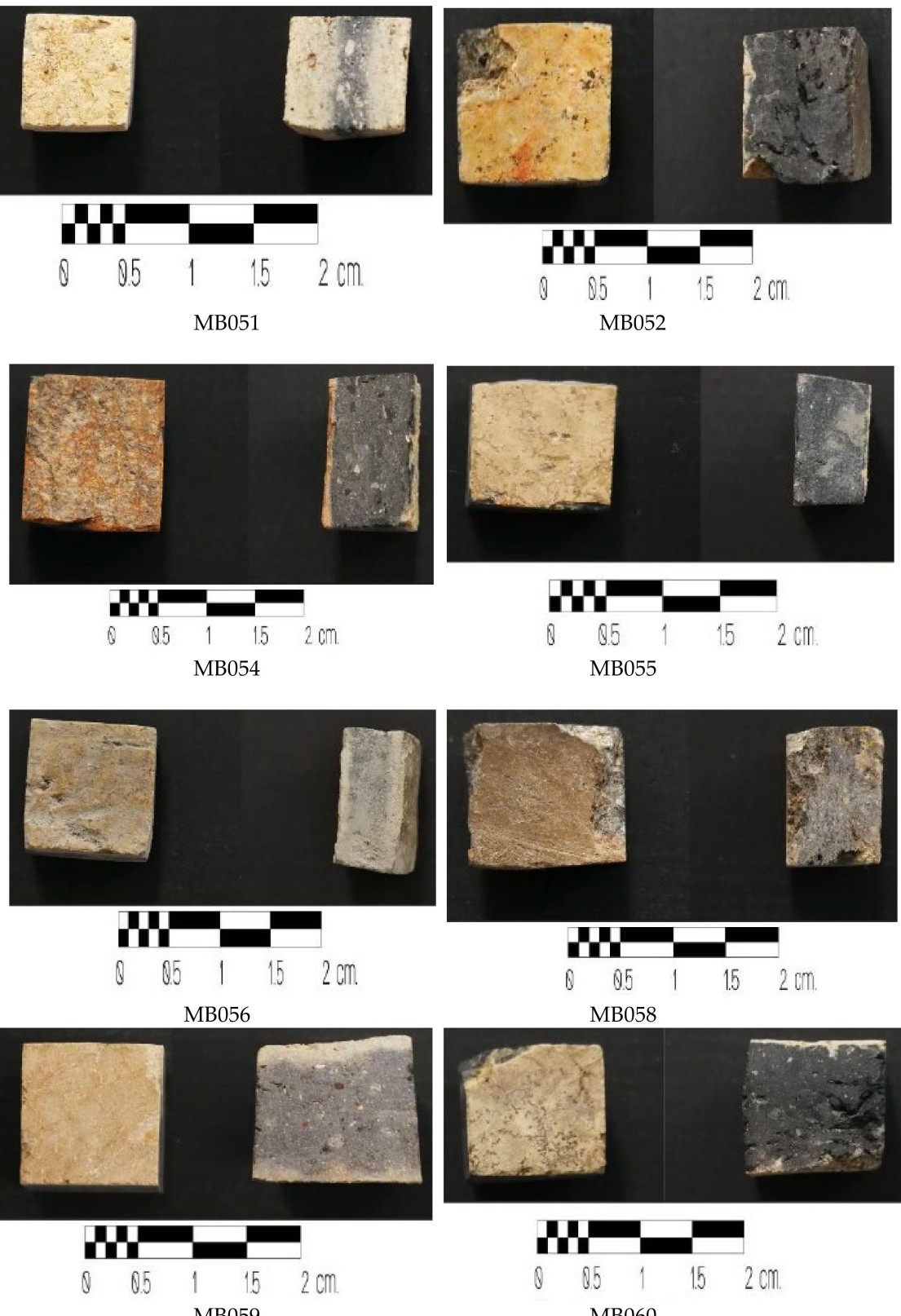

**Figure 3.** Top view (**left**) and cross-sectional (**right**) images of MB051, MB052, MB054, MB055, MB056, MB058, MB059, and MB060.

Optical glasses with known chemical compositions were used for calibration of PIXE and EDS results. It was found that the relative errors of results of two techniques were less

than 10%. PIXE results were therefore compared to EDS and the average content of each element from two techniques was evaluated [36].

Synchrotron radiation X-ray tomographic microscopy (SR XTM) was operated at XTM beamline (BL1.2W), SLRI (Nakhon Ratchasima, Thailand). For a complete dataset, X-ray projectors of each specimen were collected for 180° with 0.1° angular increment. In order to minimize the scattering and beam hardening artifacts, polychromatic X-rays were attenuated with 400 µm thick aluminum foil to allow penetration of the X-ray with mean energy of 11.5 keV for X-ray imaging. The X-ray images were taken by a 2560 × 2160 pixels CMOS camera (PCO.EDGE 5.5, Optique Peter, Lentilly, France) with a 0.7 micron resolution. The data were pre-processed and reconstructed in 3D based on a filtered-back projection algorithm using Octopus Reconstruction software (Octopus 8.9.4, TESCAN-XRE, Ghent, Belgium). The 3D projection and analysis of pores and grains were determined using Octopus Analysis software (Octopus 8.9.4, TESCAN-XRE, Ghent, Belgium). This technique was applied to examine microstructures of materials in our previous works [20,37]. The samples of MB051, MB055, MB056, and MB059 were chosen to be analyzed by this technique.

## 3. Results and Discussion

The elemental composition obtained by SEM-EDS is given in Table 2. It shows that C, O, Al, Si, K, Ca, Ti, and Fe were detected in every sample, while Na, Mg, S, Cl, and Zn were present in some samples. The average composition is given in Figure 4. The contents of trace elements analyzed by PIXE including Sc, V, Cr, Mn, Co, Ni, Cu, Zn, Ga, and Zr are shown in Table 3.

**Table 2.** Elemental composition of the burial potsherd samples collected from Ban Muang Bua probed by SEM-EDS in (a) outer layer, (b) core, and (c) inner layer.

| Sample | Layer | Elemental Composition (wt%) | | | | | | | | | | | | |
|---|---|---|---|---|---|---|---|---|---|---|---|---|---|---|
| | | C | O | Na | Mg | Al | Si | S | P | Cl | K | Ca | Ti | Fe |
| MB051 | a | 2.41 | 50.30 | 0.13 | 0.14 | 7.03 | 37.19 | N/A | 0.29 | N/A | 0.25 | 0.38 | 0.40 | 1.47 |
| | b | 3.21 | 51.14 | N/A | 0.12 | 5.76 | 37.11 | N/A | 0.64 | N/A | 0.24 | 0.43 | 0.34 | 1.01 |
| | c | 7.03 | 49.54 | 0.16 | 0.15 | 6.59 | 34.22 | N/A | 0.21 | N/A | 0.23 | 0.28 | 0.38 | 1.20 |
| MB052 | a | 5.59 | 54.60 | N/A | 0.12 | 7.17 | 27.10 | N/A | 0.15 | N/A | 0.20 | 0.58 | 0.40 | 3.70 |
| | b | 2.88 | 53.07 | N/A | N/A | 7.06 | 32.92 | N/A | 0.53 | N/A | 0.24 | 0.74 | 0.45 | 1.57 |
| | c | 5.60 | 52.42 | 0.12 | N/A | 7.22 | 31.93 | N/A | 0.40 | N/A | 0.18 | 0.67 | 0.42 | 1.04 |
| MB054 | a | 1.30 | 52.72 | 0.15 | N/A | 5.92 | 36.39 | N/A | 0.46 | N/A | 0.51 | 0.94 | 0.43 | 1.20 |
| | b | 3.49 | 52.08 | 0.11 | 0.08 | 5.97 | 36.04 | N/A | 0.15 | N/A | 0.32 | 0.77 | 0.22 | 0.76 |
| | c | 10.08 | 48.26 | 0.26 | 0.22 | 5.91 | 29.17 | N/A | 0.11 | N/A | 1.06 | 0.74 | 0.26 | 0.92 |
| MB055 | a | 1.73 | 49.25 | 0.14 | N/A | 4.34 | 41.63 | N/A | 0.30 | N/A | 0.79 | 0.60 | 0.31 | 0.90 |
| | b | 2.50 | 50.50 | 0.19 | 0.10 | 5.07 | 38.81 | N/A | 0.31 | N/A | 0.69 | 0.60 | 0.26 | 0.97 |
| | c | 3.73 | 51.95 | 0.18 | N/A | 5.55 | 36.08 | N/A | 0.24 | N/A | 0.61 | 0.53 | 0.24 | 0.89 |
| MB056 | a | 4.05 | 55.28 | N/A | N/A | 6.20 | 25.21 | N/A | 3.78 | N/A | 0.37 | 1.15 | 0.31 | 3.65 |
| | b | 3.41 | 51.06 | N/A | N/A | 5.27 | 36.95 | N/A | 0.75 | N/A | 0.33 | 0.63 | 0.36 | 1.24 |
| | c | 8.69 | 49.28 | 0.13 | 0.16 | 5.03 | 32.54 | N/A | 1.28 | N/A | 0.43 | 0.75 | 0.34 | 1.38 |
| MB058 | a | 5.15 | 51.86 | 0.12 | N/A | 6.16 | 33.33 | N/A | 0.46 | N/A | 0.50 | 0.84 | 0.57 | 1.01 |
| | b | 3.02 | 51.19 | N/A | N/A | 6.11 | 36.63 | N/A | 0.28 | N/A | 0.42 | 0.70 | 0.49 | 1.15 |
| | c | 14.33 | 46.06 | 0.11 | 0.12 | 6.37 | 28.89 | N/A | 0.95 | 0.13 | 0.49 | 0.83 | 0.56 | 1.16 |
| MB059 | a | 5.92 | 52.86 | N/A | 0.15 | 4.95 | 32.20 | N/A | 0.63 | 0.09 | 0.55 | 1.68 | 0.34 | 1.25 |
| | b | 1.92 | 50.96 | N/A | 0.12 | 5.16 | 39.18 | N/A | 0.18 | N/A | 0.56 | 0.69 | 0.34 | 0.89 |
| | c | 2.45 | 51.31 | 0.14 | 0.11 | 4.70 | 38.27 | N/A | 0.57 | N/A | 0.50 | 0.83 | 0.30 | 0.81 |
| MB060 | a | 2.95 | 53.74 | N/A | 0.20 | 7.21 | 32.77 | 1.11 | N/A | 0.11 | 0.46 | 0.85 | 0.27 | 1.34 |
| | b | 1.21 | 52.20 | 0.12 | 0.14 | 7.22 | 36.03 | N/A | N/A | N/A | 0.59 | 0.59 | 0.33 | 1.07 |
| | c | 10.82 | 46.47 | N/A | 0.19 | 3.87 | 35.56 | N/A | 0.32 | 0.09 | 0.28 | 1.02 | 0.33 | 0.55 |
| Average | a | 3.64 | 52.58 | <0.14 | <0.16 | 6.13 | 33.23 | <1.11 | <0.87 | <0.10 | 0.46 | 0.88 | 0.38 | 1.82 |
| | b | 2.35 | 51.79 | <0.14 | <0.12 | 6.15 | 36.05 | <0.10 | <0.41 | <0.10 | 0.47 | 0.60 | 0.36 | 1.17 |
| | c | 7.85 | 49.42 | <0.16 | <0.14 | 5.66 | 33.34 | <0.10 | 0.51 | <0.11 | 0.48 | 0.71 | 0.36 | 1.00 |

N/A denotes no detection of element or beyond limit of detection of SEM-EDS (<0.10 wt%).

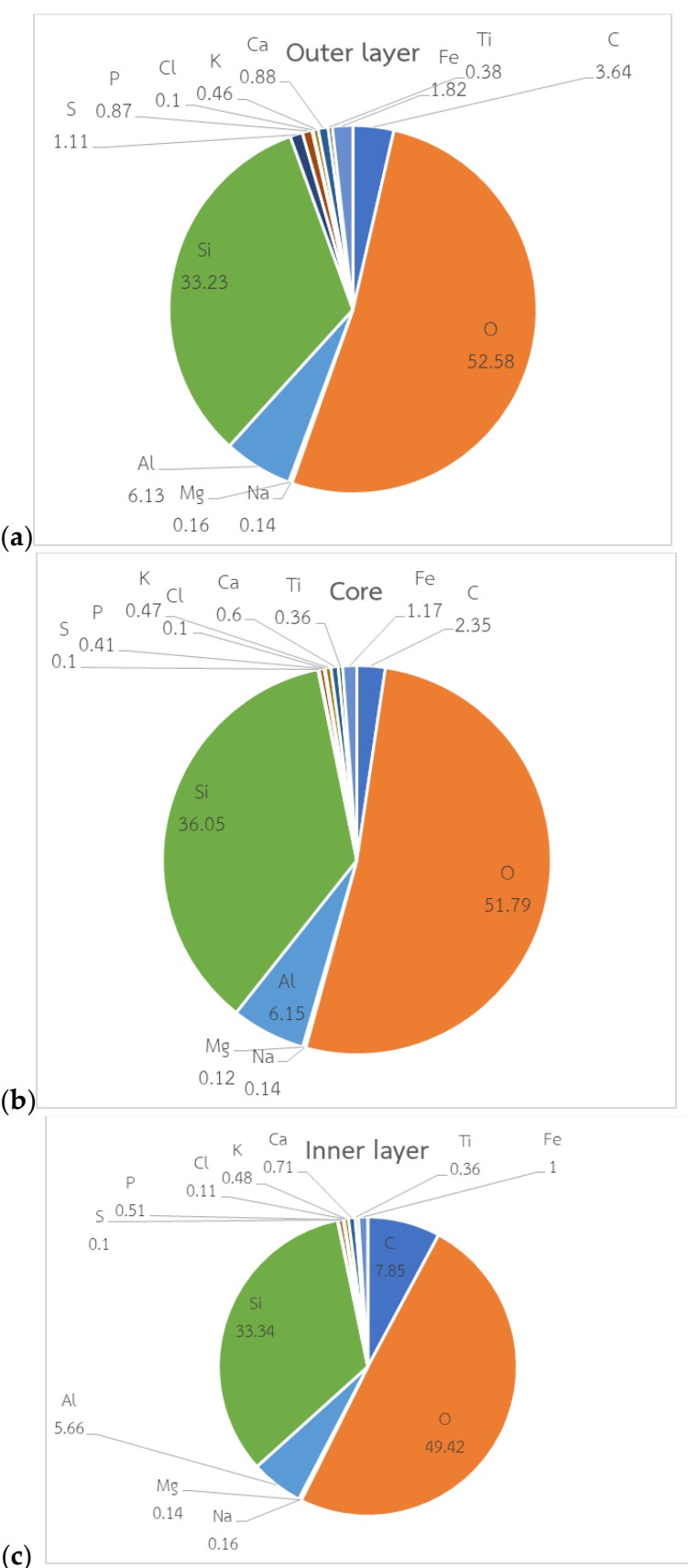

**Figure 4.** Average elemental composition of samples analyzed by SEM-EDS for (**a**) outer layer, (**b**) core, and (**c**) inner layer.

**Table 3.** Elemental composition of the burial potsherd samples collected from Ban Muang Bua probed by PIXE in (a) outer layer, (b) core, and (c) inner layer.

| Sample | Layer | Content of Element (ppm) | | | | | | | | | |
|--------|-------|------|------|------|------|------|------|------|------|------|------|
| | | Sc | V | Cr | Mn | Co | Ni | Cu | Zn | Ga | Zr |
| MB051 | a | N/A | N/A | N/A | 2.9 | N/A | N/A | N/A | N/A | 22.8 | 89.1 |
| | b | N/A | N/A | N/A | N/A | N/A | N/A | N/A | 7.3 | N/A | N/A |
| | c | N/A | N/A | N/A | N/A | N/A | N/A | N/A | 8.7 | 12.2 | N/A |
| MB052 | a | 13.2 | 13.6 | N/A | 20.0 | 27.1 | N/A | N/A | 14.6 | N/A | N/A |
| | b | N/A | 5.5 | N/A | 2.5 | N/A | N/A | N/A | 6.9 | 9.3 | N/A |
| | c | N/A | N/A | N/A | 18.7 | N/A | 9.1 | 8.7 | 23.1 | N/A | N/A |
| MB054 | a | 77.7 | N/A | N/A | 35.4 | 28.6 | N/A | N/A | N/A | 9.3 | N/A |
| | b | 12.6 | N/A | N/A | 25.9 | N/A | N/A | N/A | 7.3 | N/A | N/A |
| | c | N/A | N/A | N/A | 73.0 | 19.0 | N/A | N/A | 6.5 | N/A | N/A |
| MB055 | a | N/A | 3.8 | N/A | N/A | N/A | N/A | N/A | N/A | N/A | N/A |
| | b | N/A | N/A | N/A | N/A | N/A | N/A | 9.9 | 10.5 | N/A | N/A |
| | c | N/A | 23.1 | 6.0 | 11.2 | 31.4 | N/A | N/A | N/A | N/A | N/A |
| MB056 | a | 8.2 | N/A | 6.8 | N/A | N/A | N/A | N/A | N/A | N/A | N/A |
| | b | 7.5 | N/A | 4.9 | N/A | N/A | 10.5 | N/A | N/A | N/A | N/A |
| | c | 12.4 | N/A | 8.1 | N/A | N/A | N/A | N/A | N/A | 7.2 | N/A |
| MB058 | a | N/A | 4.8 | N/A | N/A | N/A | 6.0 | 6.7 | 7.9 | N/A | N/A |
| | b | N/A | N/A | 6.7 | 29.6 | N/A | 11.3 | N/A | 12.2 | N/A | N/A |
| | c | N/A | N/A | N/A | N/A | N/A | N/A | N/A | 9.5 | N/A | N/A |
| MB059 | a | 3.1 | N/A | N/A | 1.8 | N/A | N/A | N/A | N/A | N/A | N/A |
| | b | N/A | 19.5 | N/A | N/A | N/A | N/A | N/A | N/A | N/A | N/A |
| | c | 16.6 | N/A | N/A | 5.8 | N/A | N/A | N/A | N/A | 7.2 | N/A |
| MB060 | a | N/A | N/A | 9.1 | 43.7 | N/A | N/A | 8.8 | 7.3 | 7.2 | N/A |
| | b | N/A | 9.6 | N/A | N/A | N/A | 10.1 | N/A | 10.3 | N/A | N/A |
| | c | N/A | N/A | N/A | 8.2 | N/A | N/A | N/A | N/A | N/A | N/A |

N/A denotes no detection of element or beyond limit of detection of PIXE (<1 ppm).

Si and Al were found as the major components with some variation of composition. The compositions were in the ranges of 27.10 to 41.63 and 3.87 to 7.22 wt%, respectively. Fe existed in all samples originated from the raw materials of earthen clays which affected the red coloration of pottery. A small amount of Fe showed the coloration from brown to orange and white depending on the complete firing process. As shown in Figure 5, the amounts of Si, Al, and Fe in a core tended to be larger, approximately the same, and lower, respectively, than outer and inner layers.

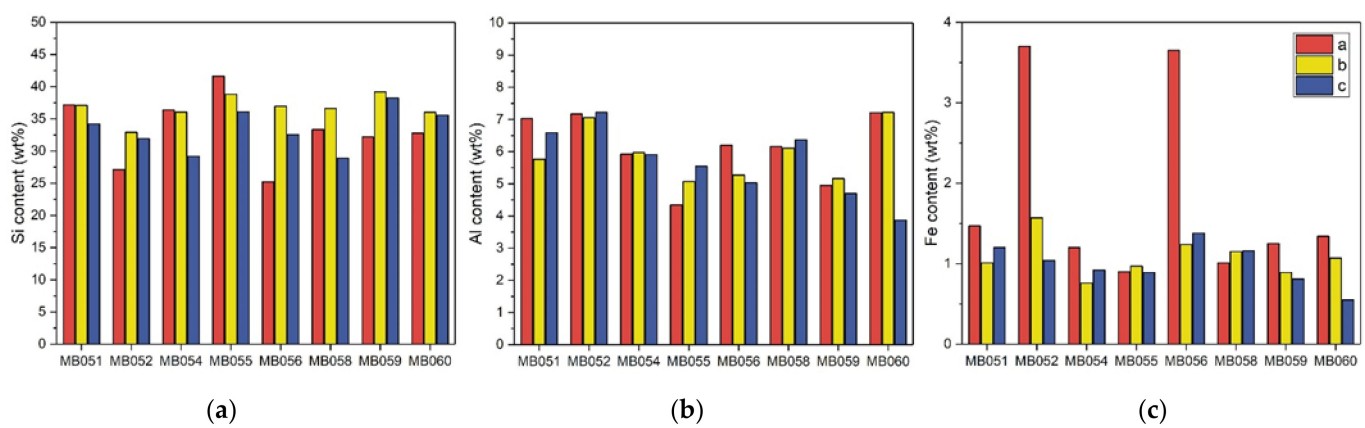

**Figure 5.** Contents of characteristic elements of Si, Al, and Fe in (a) outer, (b) core, and (c) inner layers of Ban Muang Bua potsherd samples.

Ca (~0.28–1.68 wt%) and K (~0.18–1.06 wt%) were also detected. Detection of other elements reflected the same area of production of potteries where the clays were procured. The low contents of Ca and K for both outer and inner layers indicated low firing temperatures for the two layers of samples (~600–900 °C) in open-air kilns [38,39]. Ca was added to the clay paste as temper [40]. It has been reported that the source of phosphorous (P) contamination in ancient potteries was different, i.e., minerals, bone ashes that were added into the clay paste, and burial environment [41–43]. P contained in all samples, especially in the inner layer, was presumed to be come from the burial environment.

Trace elements such as Sc, V, Cr, Mn, Co, Ni, Cu, Zn, Ga, and Zr were detected using PIXE due to their contents beyond limit of detection of SEM-EDS. The contents (ppm) of Sc (3.1–77.7), V (3.8–19.5), Cr (4.9–9.1), Mn (1.8–73.0), Co (19.0–28.6), Ni (6.0–11.3), Cu (6.7–9.9), Zn (6.5–60.3), Ga (7.2–22.8), and Zr (89.1) are shown in Table 3. Mn (MB056), Zn (MB056 and MB058), and Ga (MB055) were below detection limit in some samples. Zr was only found in MB051, while Cr (MB056, MB058, and MB060), Co (MB052, MB054, and MB055), Ni (MB052, MB056, MB058, and MB060), and Cu (MB052, MB055, MB058, and MB060) were detected in some samples. Sc (MB052, MB054, MB056, and MB059) and V (MB052, MB055, MB058, MB059, and MB060) were present in some samples. These elements were trace elements in soil and clastic sedimentary rocks found in northeastern Thailand [44].

The variation of elemental composition indicated different characteristics of raw materials. Geological data indicated that these raw materials were sandy soil derived from marine sands, clays, and salt deposits that were mostly iron-rich-kaolin clay [44]. This archaeological evidence reflected iron smelting and salt manufacturing in this area [45,46].

Raw materials with a high proportion of Si possessed low mineral contaminations. The possession resulted in lower plasticity, unfired strength, and drying shrinkage than one with the high mineral contents. However, large amount of Al was also able to reduce the plasticity [39,47].

SEM micrographs of samples MB051, MB052, MB054, MB055, MB056, MB058, MB059, and MB060 are shown in Figure 6. The figure shows the different microstructure and morphology between core, and outer and inner layers for each sample. It is obvious that the core contained heterogeneous morphologies of composites in irregular plate-like particles with a wide range of size distribution while the outer and inner layers possessed slipped microstructure of glazes. The core showed none or only a small amount of the glass phase. This difference was due to the phase transformation of raw material compounds during the firing process.

Figure 7 shows the SR XTM 3D reconstructive results which provided information about the porosity of the samples. The results in Table 4 exhibit some variation of porosity between a core, and outer and inner layers of four selected samples of MB051, MB055, MB056, and MB059. The total porosity of a core tended to be higher than the outer and inner layers. Some samples possessed high porosity in outer and inner layers due to possession of thin layers in which porosity from some parts of core were combined. The results corresponded to SEM in which outer and inner layers possessed dense microstructure of glazes, while the core contained a small amount of the dense glass phase.

Even though the pottery samples in this work were grave goods excavated from the location nearby our previous work which was the pottery used in daily life [20], the results in Table 5 show that their elemental compositions were slightly different. It is proposed that their raw materials originated from the same source. The potteries in our previous and present works might be produced with different kilns and amounts of raw materials but fired with open-air kilns at low temperatures (<900 °C). The small difference in composition resulted from variations of the production or the use of raw materials from different batches as well as weathering and surface contamination.

It is proposed that Thung Kula Rong Hai had its unique culture compared to the contemporaneous cultures in northeastern Thailand. Pottery in each burial site possessed unique characteristics and are the main source of information about the way of life dur-

ing the late prehistory in this area. Investigation of other samples in this site for more information will be carried out for future work.

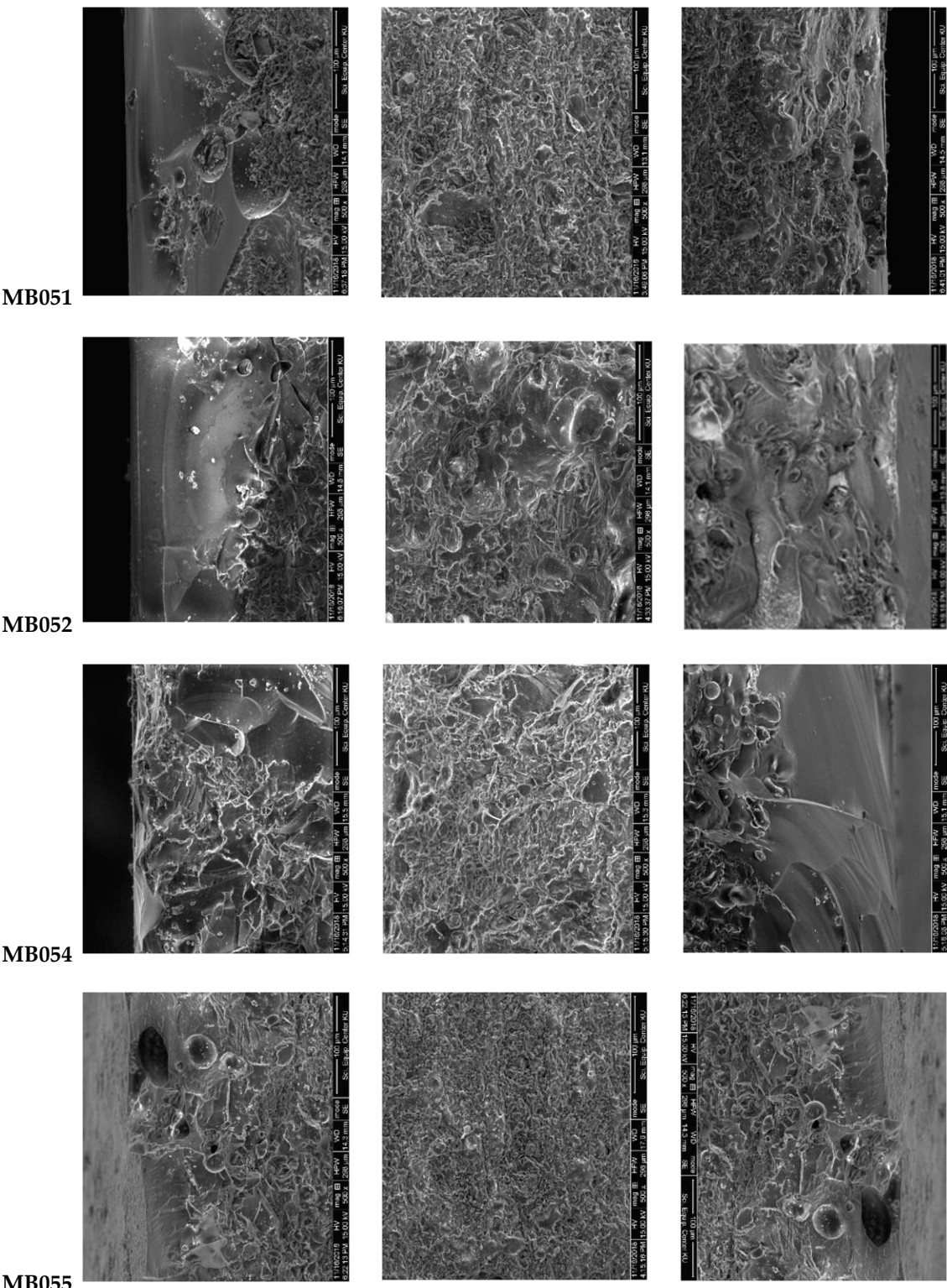

**MB051**

**MB052**

**MB054**

**MB055**

**Figure 6.** *Cont.*

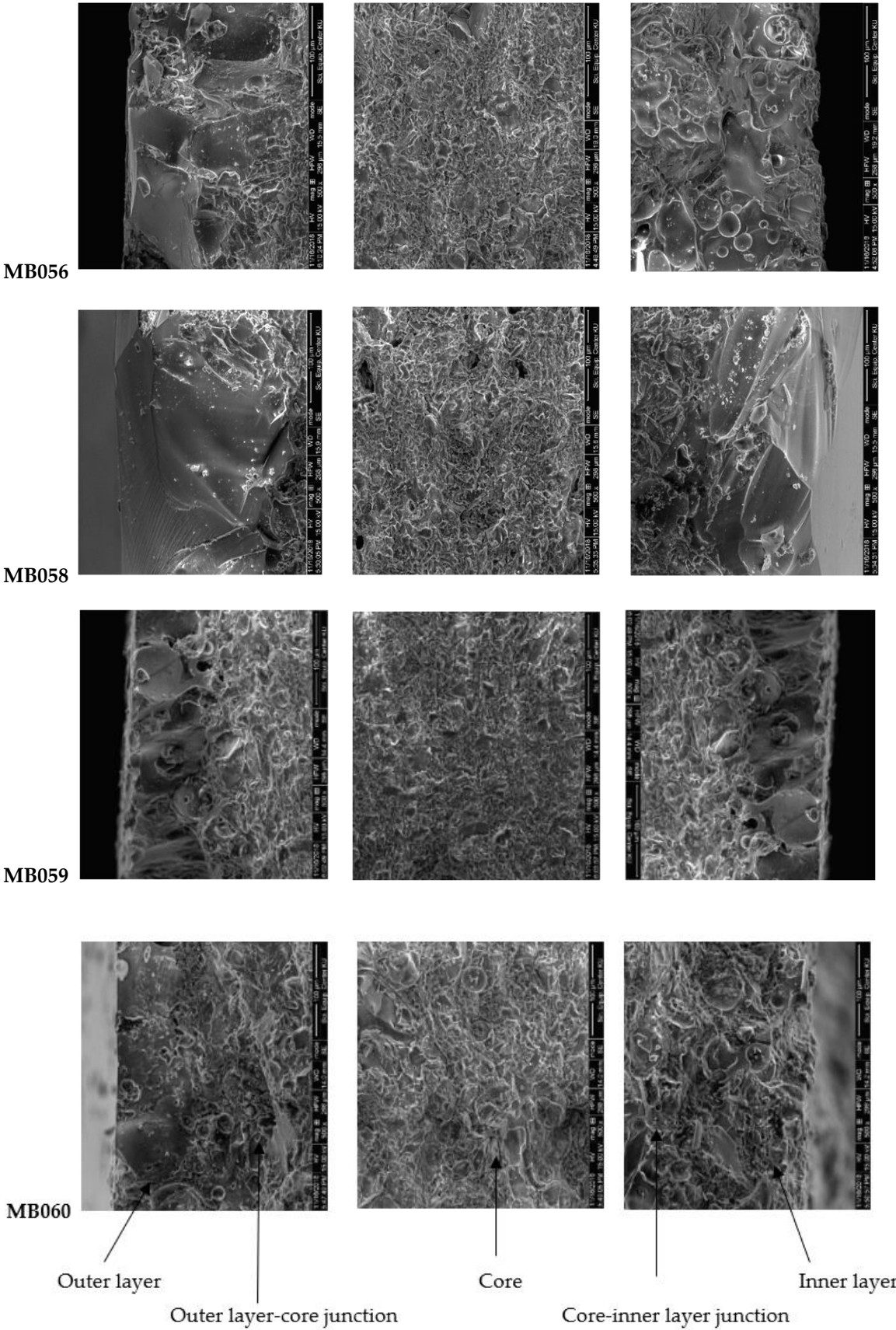

**Figure 6.** SEM micrographs of outer layer, core, and inner layer of MB051, MB052, MB054, MB055, MB056, MB058, MB059, and MB060.

**Table 4.** Porosities of 4 selected samples analyzed using SR XTM.

| Sample | Side | Porosity (%) | | |
|---|---|---|---|---|
| | | Open | Closed | Total |
| MB051 (Very thin inner layer) | Outer layer | 15.1 | 0.6 | 15.7 |
| | Core | 9.3 | 1.0 | 10.3 |
| | Inner layer | 11.6 | 0.7 | 12.4 |
| MB055 (Very thin outer and inner layers) | Outer layer–core junction | 5.3 | 1.0 | 6.2 |
| | Core | 4.5 | 0.8 | 5.0 |
| | Core–inner layer junction | 1.0 | 1.2 | 2.1 |
| MB056 (Thin outer and inner layers) | Outer layer | 1.6 | 1.1 | 2.7 |
| | Core | 1.9 | 0.8 | 2.7 |
| | Core–inner layer | 2.5 | 0.9 | 3.3 |
| | Inner layer | 0.5 | 0.8 | 1.2 |
| MB059 (Thin outer and inner layers) | Outer–core junction | 1.4 | 0.6 | 2.0 |
| | Core | 5.6 | 1.4 | 6.0 |
| | Core–inner junction | 1.0 | 0.8 | 1.8 |
| | Inner layer | 0.1 | 0.2 | 0.3 |

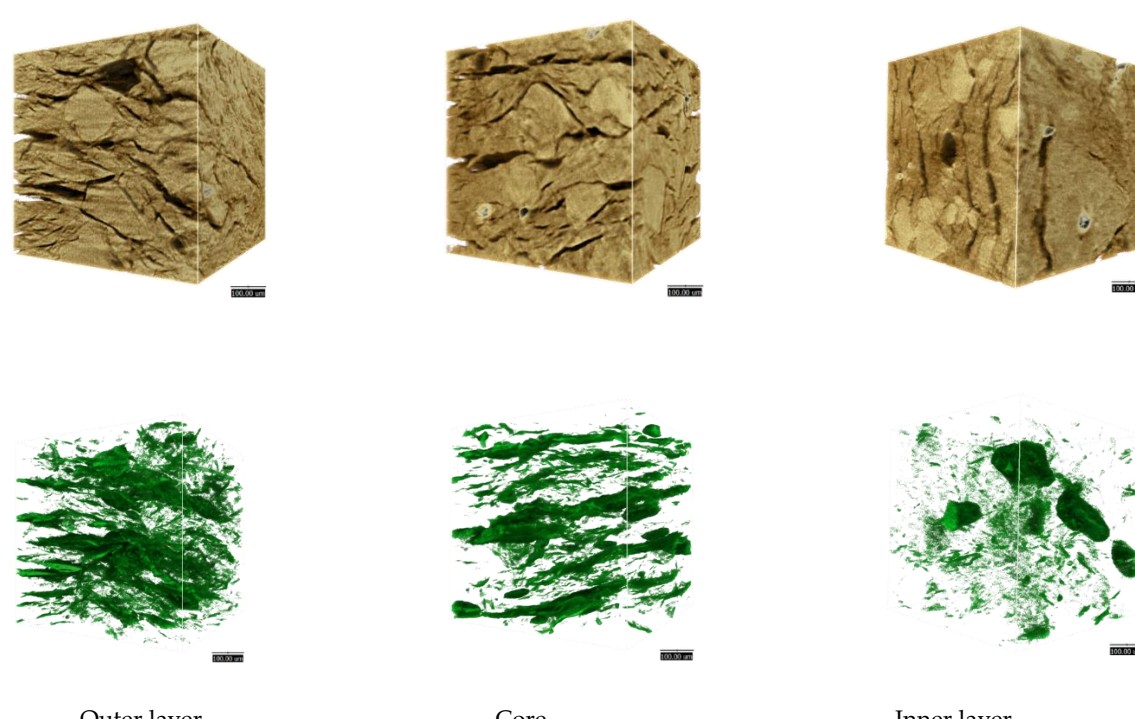

MB051          Outer layer                    Core                          Inner layer

**Figure 7.** *Cont.*

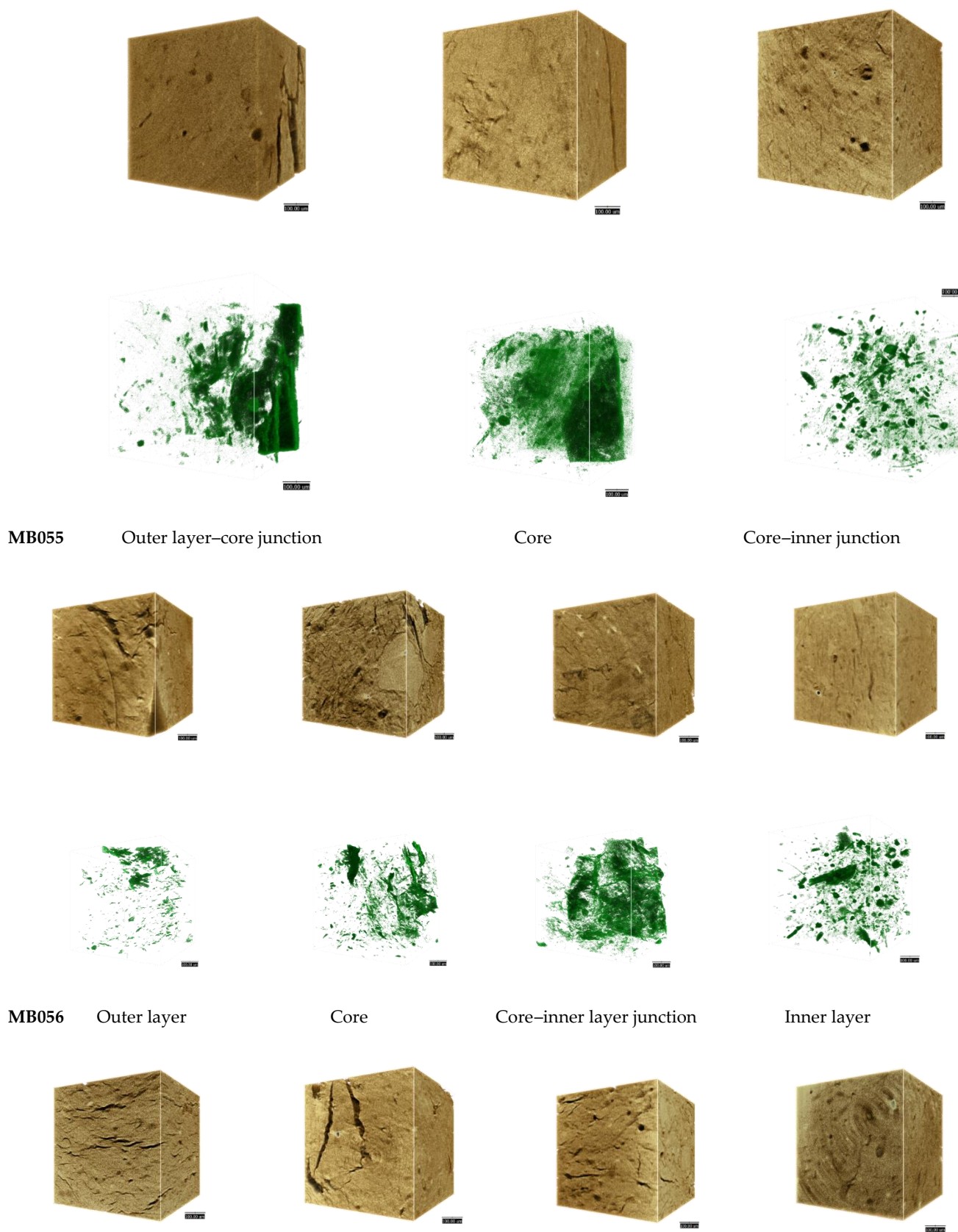

**Figure 7.** *Cont.*

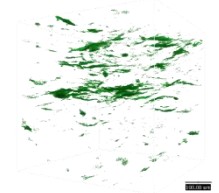 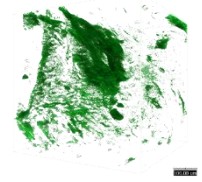 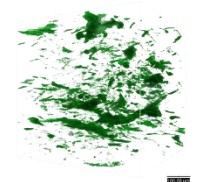 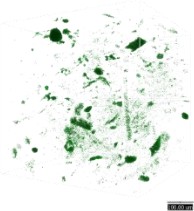

| MB059 | Outer layer–core junction | Core | Core–inner layer junction | Inner layer |

**Figure 7.** Tomographic images of MB051, MB055, MB056, and MB059 which provided visualization of internal structure (brown), and pore and crack (green). The length of scale bars located at the bottom right corner or top right corner of images is 100.00 μm.

**Table 5.** Comparison between average elemental compositions of Ban Muang Bua pottery in previous (Data from [20]) and present works. Adapted with permission from Nimsuwan, N.; Dutchaneephet, J.; Pakawanit, P.; Dararutana, P.; Won-in, K. (2019) (Malays. J. Microsc.) [20].

| Sample | Elemental Composition (wt%) | | | | | | | | | | | | |
|---|---|---|---|---|---|---|---|---|---|---|---|---|---|
| | **C** | **O** | **Na** | **Mg** | **Al** | **Si** | **S** | **P** | **Cl** | **K** | **Ca** | **Ti** | **Fe** |
| Outer layer | | | | | | | | | | | | | |
| Previous | 4.66 | 52.14 | <0.17 | <0.23 | 5.00 | 35.19 | <0.10 | <1.04 | <0.10 | 0.29 | <0.41 | <0.34 | 1.01 |
| Present | 3.64 | 52.58 | <0.14 | <0.16 | 6.13 | 33.23 | <1.11 | <0.87 | <0.10 | 0.46 | 0.88 | 0.38 | 1.82 |
| Core Previous | 3.99 | 52.19 | 0.12 | <0.10 | 5.68 | 35.05 | <0.10 | 1.04 | <0.10 | 0.30 | 0.44 | 0.33 | 1.15 |
| Present | 2.35 | 51.79 | <0.14 | <0.12 | 6.15 | 36.05 | <0.10 | <0.41 | <0.10 | 0.47 | 0.60 | 0.36 | 1.17 |
| Inner layer | | | | | | | | | | | | | |
| Previous | 5.68 | 51.20 | <0.28 | <0.13 | 6.27 | 32.74 | <0.10 | 1.20 | <0.20 | 0.36 | 0.60 | 0.46 | 1.35 |
| Present | 7.85 | 49.42 | <0.16 | <0.14 | 5.66 | 33.34 | <0.10 | 0.51 | <0.11 | 0.48 | 0.71 | 0.36 | 1.00 |

## 4. Conclusions

X-ray spectroscopic characterization of the ancient burial potteries produced at Ban Muang Bua archaeological site, Roi Et province (northeastern Thailand) was achieved. The amounts of Si, Al, and Fe in core tended to be larger, approximately the same, and lower, respectively, than outer and inner layers.

SEM micrographs showed morphologies of composites in the core that contained irregular plate-like particles with a wide range of size distribution, and the dense outer and inner layers of glazes. The results corresponded to SR XTM which revealed high porosity of the core. The low contents of Ca and K for both outer and inner layers indicated low firing temperatures of samples (~600–900 °C) in open-air kilns. These results implied the production technology of ancient pottery.

The raw materials of pottery in our present and previous works were originated from the same source. The two pottery types might be produced with different kilns and amounts of raw materials. The raw materials were sandy soil derived from marine sands, clays, and salt deposits that were mostly iron-rich-kaolin clay. This archaeological evidence reflected iron smelting and salt manufacturing in this area.

**Author Contributions:** Conceptualization, K.W.-i., C.B., and P.D.; methodology, K.W.-i. and P.D.; software, C.T. and P.P.; validation, C.B., K.W.-i., P.P., U.T., and P.D.; formal analysis, N.N., C.T., P. P., and P.D.; investigation, C.B., K.W.-i., N.N., P.P., U.T., and P.D.; resources, K.W.-i. and N.N.; data curation, C.B., K.W.-i., and P.D.; writing-original draft preparation, K.W.-i. and P.D.; writing-review and editing, C.B., K.W.-i., P.P., U.T., and P.D.; visualization, N.N., P.P., and P.D.; supervision, K.W.-i. and C.B.; project administration, K.W.-i.; funding acquisition, K.W.-i., C.B. and U.T. All authors have read and agreed to publish version of the manuscript.

**Funding:** This research is partly funded by the Faculty of Science (Department of Earth Science) at Kasetsart University and Center of Excellence in Materials Science and Technology (Chiang Mai University). In-kind contribution in terms of the uses of laboratory, equipment, chemicals, and operators is supported by Plasma and Beam Physics Research Facility (Department of Physics and Materials Science, Faculty of Science, Chiang Mai University, Chiang Mai, Thailand) and Department of Earth Sciences (Faculty of Science, Kasetsart University, Bangkok, Thailand).

**Data Availability Statement:** All data generated or analyzed during this study are included in this article.

**Acknowledgments:** Authors kindly thank the 11th Regional Office of Fine Arts in Ubon Ratchathani province for providing the samples. BL1.2W at SLRI (Nakhon Ratchasima, Thailand), Science Equipment Center at Faculty of Science (Kasetsart University, Bangkok, Thailand), and Plasma and Beam Physics Research Facility at Chiang Mai University (Department of Physics and Materials Science, Faculty of Science) are thanked for providing SR XTM, SEM-EDS, and PIXE facilities, respectively. Pichai Sirisangsawang, Purin Prommaban and Kumaree Thongimboon, Sudarat Wongke, and the BL1.2W operator team are also thanked for SEM-EDS operation, PIXE operation, PIXE analysis, and SR XTM analysis, respectively.

**Conflicts of Interest:** The authors declare no conflict of interest. The funders had no role in the design of the study, including the collection, analyses, interpretation of data; in the writing of this manuscript and the decision to publish the results.

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
