# Peer review of "Characterization of Ancient Burial Pottery of Ban Muang Bua Archaeological Site (Northeastern Thailand) Using X-ray Spectroscopies"

_applsci, doi:10.3390/app12052568_

Round 1

Reviewer 1 Report

I do not see a research question or hypotheses given for this submission, regarding production/trade of archaeological ceramics. The archaeological site apparently spans 2000 years, but what time period(s) are the samples from? What has been learned from other studies in this region? One reference is given for previous work by most of the authors, but the data are not combined/compared.

The number of samples (8) is very small. To address where the clay is coming from, there should also be some clay samples tested, and discussion of data from other studies in Thailand. There should be a significant section on "previous research" addressing pottery trade in this region, and another on selection of analytical methods and sample selection to address hypotheses about trade in the time period represented.

While analytical methods were applied, SEM-EDS only measures major/minor elements, and pottery trade/sourcing studies need analyses that include trace elements (e.g. INAA or ICP-MS). What was learned from the 3D analysis?

The references cited appear to be a mixed collection, with very few on archaeological ceramics. Major journals with published articles on ceramic analyses include Journal of Archaeological Science, Archaeometry, and several others.

Author Response

Point 1: I do not see a research question or hypotheses given for this submission, regarding production/trade of archaeological ceramics. The archaeological site apparently spans 2000 years, but what time period(s) are the samples from? What has been learned from other studies in this region? One reference is given for previous work by most of the authors, but the data are not combined/compared.

Response 1: The aim of this work has been clarified and addressed (green highlight) in the last paragraph of Introduction. The dating information of samples and relationship with other ancient burial potteries have been specified (turquoise highlight) in Introduction. Data comparison and discussion with previous work have been added in the last two paragraphs (green highlight) in Discussion.

Point 2: The number of samples (8) is very small. To address where the clay is coming from, there should also be some clay samples tested, and discussion of data from other studies in Thailand. There should be a significant section on "previous research" addressing pottery trade in this region, and another on selection of analytical methods and sample selection to address hypotheses about trade in the time period represented.

Response 2: Adding of PIXE results of 8 samples combined with previous geological research can provide adequate information about trace elements which implied the source of clay as marked by turquoise highlight in Discussion. As the aim of this work has been clarified that to compare the composition to pottery from our previous work that can lead to the information about raw materials and production technology of the pottery in this work (marked by green highlight in the last paragraph of Introduction), the issue about pottery trade has been removed.

Point 3: While analytical methods were applied, SEM-EDS only measures major/minor elements, and pottery trade/sourcing studies need analyses that include trace elements (e.g. INAA or ICP-MS). What was learned from the 3D analysis?

Response 3: The analysis of trace element has been done using PIXE and the results and discussion have been added as shown in Table 3 and the contents marked by turquoise highlight in Discussion. Additional results of 3D tomographic images (Figure4) provide visualization of internal structure and information including porosity of samples as the contents marked by pink highlights in Results and Discussion.

Point 4: The references cited appear to be a mixed collection, with very few on archaeological ceramics. Major journals with published articles on ceramic analyses include Journal of Archaeological Science, Archaeometry, and several others.

Response 4: This works contains adequate references about archaeological ceramics including references No. 5-26.

Note: Additional contents in article which have not been directly mentioned by reviewers are marked by gray highlights.

Reviewer 2 Report

The paper presents a study about ancient potteries coming from Ban Muang Bua via SEM-EDX and SR XTM. Data coming from the different techniques are used together to characterize the objects. Although the paper is really interesting, the results, discussion and conclusions could be improved revise the text with more details together with the impact of this research. 

Author Response

Response to Reviewer 2 Comments

Point 1: The paper presents a study about ancient potteries coming from Ban Muang Bua via SEM-EDX and SR XTM. Data coming from the different techniques are used together to characterize the objects. Although the paper is really interesting, the results, discussion and conclusions could be improved revise the text with more details together with the impact of this research.

Response 1: The Results, Discussion, and Conclusions have been revised (marked by color highlights in Results, Discussion, and Conclusions).

Note: Additional contents in article which have not been directly mentioned by reviewers are marked by gray highlights.

Reviewer 3 Report

Remarks in added pdf

Author Response

Response to Reviewer 3 Comments

Point 1: Page 1 Line 3-4: X-ray spectroscopy; Only one technique specify in the title, change the range of title

Response 1: The word “spectroscopy” in title of article has been changed to “spectroscopies” (green highlight) for correspondence with the techniques used in this study.   

Point 2: Page 1 Line 19-21; Another technique of pots examination

Response 2: Additional information about analytical technique of PIXE has been addressed in Abstract (turquoise highlight).

Point 3: Page 1 Line 24: origin of potsherd; Origin of pots? Source of raw materials

Response 3: The sentence has been revised and the importance of combined techniques has been specifically described (green highlights in Abstract).

Point 4: Page 1 Line 26: archaeological research; too large scale of archaeological works

Response 4: The sentence has been revised and the importance of combined techniques has been specifically described (green highlights in Abstract).

Point 5: Page 2 Figure 1; Correct the figure, resolution of foto and legibling of maps

Response 5: Figure 1 has been revised.

Point 6: Page 2 Line 58; Add age of pots, culture, etc.

Response 6: Information about the age of pottery and culture have been added in Introduction (pink highlight).

Point 7:  Page 2 Line 66-69; To introduce add: question of the research, goals, and solving of problem.

Response 7: The aim of this work has been clarified and addressed (green highlight) in the last paragraph of Introduction.

Point 8: Page 2 Line 71 Samples; Add table with description of samples, pots as a archaeological materials

Response 8: The descriptions of samples have been added and resolution of images has been improved as shown in Table 1.

Point 9:  Page 3 Figure 2; Correct the Figure: resolution of foto

Response 9: The descriptions of samples have been added and resolution of images has been improved as shown in Table 1.

Point 10:  Page 3 Line 86; No in result

Response 10: Additional content and images of SEM have been added to Discussion (marked by dark yellow highlight).

Point 11:  Page 3 Line 99; No exist in results

Response 11: The content has been added as marked by pink highlight in Discussion.

Point 12: Page 3 Line 101; Add another samples of research characteristics and documentation foto

Response 12: Additional content and images of SEM have been added to Discussion (marked by dark yellow highlight).

Point 13: Page 3 Line 107-110; This is interpretation, change place of this fragment

Response 13: The content has been moved to Discussion (2nd paragraph of turquoise highlight).

Point 14: Page 3 Line 111-112; It is not results, this is methodology, change place of it

Response 14: The content has been improved for being results (pink highlight in Results).

Point 15: Page 4 Figure 4; Correct resolution

Response 15: The quality of Figure 2 (former Figure 4) has been improved.

Point 16: Page 4 Figure 5; Add symbols of each part of Figure, and explain it. Add another samples of pots and compare properties: pore, cracks, etc.

Response 16: Additional images and content have been added (pink highlight in Discussion).

Point 17: Page 5 Table 1; Explain a, b and c

Response 17: The meanings of a, b, and c have been added to the caption of Table 2 (former Table 1).

Point 18: Page 5 Discussion; Add graphs of difference layers and core

Response 18: The graphs of contents of characteristic elements in outer (a), core (b), and inner layers (c) of samples have been added as shown in Figure 5.

Point 19: Page 6 Line 179; Not in results

Response 19: The content exists in Results (dark yellow highlight).

Point 20: Page 6 Line 180; No in discussion

Response 20: Additional content has been added to the 3rd paragraph (green highlight) of Discussion.

Point 21: Page 6 Line 185; No in discussion

Response 21: This content has been removed and new contents have been added for agreement with the aim of this study (green highlight in Conclusion).

Note: Additional contents in article which have not been directly mentioned by reviewers are marked by gray highlights.

Round 2

Reviewer 1 Report

You have made noticeable improvements to your prior submission. My main concerns are still about the small number of artifacts tested and how we can be sure these are representative of the much larger excavated collection.

The radiocarbon dates from 2002-2003 (lines 85-86) have such large calibrated age ranges that they are virtually useless. If including  them, at least acknowledge this and how the TL dates are much more precise.

What are "optical glasses" (line 128)

Throughout, it is important to use "SEM-EDS" and not just "EDS", since EDS can also be with XRF and other instruments. Lines 177, 195, 246, 331.

On lines 170-171, make it clear that the Ca and K values are lower for the inner/outer layers, with the low firing temp not reaching the interior (core).

In paragraph lines 178-183 make it clear that certain elements are "below detection limit", rather than "not present".

In discussing the results regarding composition, make it clear whether you are referring to major elements (kind of clay) or trace elements (sources). Most pottery sourcing studies use the incompatible elements (Rb, Sr, Y, Zr, Nb) which apparently you are not getting from PIXE.

Minor corrections needed:

line 4: change "Spectroscopies" to "Spectroscopy" in the title
line 30: change "potsherd" to "pottery"
line 35: add space before "C"
line 64: delete "to of between" and add "from the" so it reads "...are dated from the late prehistoric period..."
line 79: change "This" to "These"
line 83: remove "s" from "evidences", add "s" to "reveal", and insert "the" before "Ban Muang Bua"
line 125: the "2" in Na2O should be subscript
line 149: remove comma after "Zr"
line 160: delete "-level"
line 166: add "a" before "core"
line 168: remove spaces before/after the dashes
line 171: add space before "C"

Table 2: check the layout for the first column, items are overlapping
Table 3: make this table like Table 2 with columns for each element

line 255: delete "of" at end of line
line 274: add "the" and an "s" to "sample" so it reads "...the porosity of the samples"
lines 276 and 278: add "a" before "core"
line 298: change "two potteries" to "two pottery types"
line 300: add space before "C"

Table 4: one decimal place for the % porosity is sufficient

line 316: change to "pottery"
lines 320 and 323: change "potteries" to "pottery"
line 332: add "the" before "ppm level."
line 339: remove spaces before/after hyphen; add space before "C"
lines 341-342: change "potteries" to "pottery" and "pottery types"

Author Response

Response to Reviewer 1 Comments

Round 2

The contents related to the reviewer comments are marked as yellow highlights in the manuscript.

Point 1: You have made noticeable improvements to your prior submission. My main concerns are still about the small number of artifacts tested and how we can be sure these are representative of the much larger excavated collection.

Response 1: These samples have the same typology which corresponds to most of the potsherds found in this archeological site.

Point 2: The radiocarbon dates from 2002-2003 (lines 85-86) have such large calibrated age ranges that they are virtually useless. If including them, at least acknowledge this and how the TL dates are much more precise.

Response 2: Radiocarbon dates (2010) and TL dates (July 30, 2020) have been specified as shown in Page 3, Lines 84 and 88.

Point 3: What are "optical glasses" (line 128)?

Response 3: Optical glass is a high purity glass with a certain composition used for production of lens.

Point 4: Throughout, it is important to use "SEM-EDS" and not just "EDS", since EDS can also be with XRF and other instruments. Lines 177, 195, 246, 331.

Response 4: The “EDS” has been changed to “SEM-EDS”; Page 6 Line 174, Page 8 Line 187, and Page 8 Line211.

Line 331, we have deleted this paragraph (Comment of Reviewer#3).

Point 5: On lines 170-171, make it clear that the Ca and K values are lower for the inner/outer layers, with the low firing temp not reaching the interior (core).

Response 5: The sentence in the 1st paragraph after Figure 5 has been revised as “The low contents of Ca and K for both outer and inner layers indicated low firing temperatures for the two layers of samples (~600-900° C) in open-air kilns [38-39]. “ for better clarification, as shown in Page 8 Line 205.

Point 6: In paragraph lines 178-183 make it clear that certain elements are "below detection limit", rather than "not present".

Response 6: The phrase in the 2nd paragraph after Figure 5 has been revised, as shown in Page 8 Line 214.

Point 7: In discussing the results regarding composition, make it clear whether you are referring to major elements (kind of clay) or trace elements (sources). Most pottery sourcing studies use the incompatible elements (Rb, Sr, Y, Zr, Nb) which apparently you are not getting from PIXE.

Response 7: Actually, examination of many trace elements had been done but the 5 elements mentioned by reviewer had not been found. Even though PIXE can analyze the element in ppm, these 5 elements were not detected, except only Zr detected in the outer layer of MB051. It is possible that these elements may be present with very small quantity in that beyond the efficiency of detector. In this work, the finding of source of raw materials was therefore focused on the geological evidence which implied that the source was sandy soil derived from marine sands, clays, and salt deposits that were mostly iron-rich-kaolin clay.

Minor corrections needed:

The corrections have been done (yellow highlight in manuscript).

Point 8: line 4: change "Spectroscopies" to "Spectroscopy" in the title

Response 8: Formerly we use “spectroscopy” in the title but Reviewer#3 implied us to change it to “spectroscopies”

Point 9: line 30: change "potsherd" to "pottery"

Response 9: We have already changed it; Page 1 Line 30.

Point 10: line 35: add space before "C"

Response 10: We have already done; Page 1 Line 35.

Point 11: line 64: delete "to of between" and add "from the" so it reads "...are dated from the late prehistoric period..."

Response 11: We have already done; Page 2 Lines 64-65.

Point 12: line 79: change "This" to "These"

Response 12: We have already done; Page 2 Line 79.

Point 13: line 83: remove "s" from "evidences", add "s" to "reveal", and insert "the" before "Ban Muang Bua"

Response 13: We have already done; Page 3 Line 83.

Point 14: line 125: the "2" in Na2O should be subscript

Response 14: We have already done: Page 6 Line 147.

Point 15: line 149: remove comma after "Zr"

Response 15: We have already done; Page 6 Line 171.

Point 16: line 160: delete "-level"

Response 16: We have already done; Page 5 Line 128.

These sentences were transferred to the first paragraph in Methods (Comment of Reviewer#3).

Point 17: line 166: add "a" before "core"

Response 17: We have already done; Page 8 Line 195.

Point 18: line 168: remove spaces before/after the dashes

Response 18: We have already done; Page 8 Line 202.

Point 19: line 171: add space before "C"

Response 19: We have already done Page 8 Line 205.

Point 20: Table 2: check the layout for the first column, items are overlapping

Response 20: We have already done; Page 6.

Point 21: Table 3: make this table like Table 2 with columns for each element

Response 21: We have already done; Page 9.

Point 22: line 255: delete "of" at end of line

Response 22: We have already done: Page 9 Lines 237-238.

Point 23: line 274: add "the" and an "s" to "sample" so it reads "...the porosity of the samples"

Response 23: We have already done; Page 12 Line 259.

Point 24: lines 276 and 278: add "a" before "core"

Response 24: We have already done; Page 12 Line 261.

Point 25: line 298: change "two potteries" to "two pottery types"

Response 25: We have already changed to “The potteries in our previous and present works” (Comment of Reviwer#3), as shown in Page 14 Lines 291-292.

Point 26: line 300: add space before "C"

Response 26: We have already done; Page 14 Line 293.

Point 27: Table 4: one decimal place for the % porosity is sufficient

Response 27: We have already done; Page 12.

Point 28: line 316: change to "pottery"

Response 28: We have already done; Page 14 Line 297.

Point 29: lines 320 and 323: change "potteries" to "pottery"

Response 29: We have already changed; Page 14 Line 302.

                         We have deleted the first paragraph (Comment of Reviewer#3).

Point 30: line 332: add "the" before "ppm level."

Response 30: We have deleted these paragraphs (Comment of Reviewer#3).

Point 31: line 339: remove spaces before/after hyphen; add space before "C"

Response 31: We have already done; Page 15 Line 316.

Point 32: lines 341-342: change "potteries" to "pottery" and "pottery types"

Response 32: We have already changed; Page 15 Lines 318-319.

Reviewer 2 Report

The paper was significantly improved and the unclear sectons was significantly revised.

Author Response

"-"

Reviewer 3 Report

comments in the text (PDF)

Author Response

Response to Reviewer 3 Comments

Round 2

The contents related to the reviewer comments are marked as green highlight in the manuscript.

Point 1: Page 3 Line 97: add documentation from archaeological field research where the samples were taken, so that the archaeologist can benefit from this

Response 1: The information has been added to the first sentence in Materials and Methods, Page 3 Lines 97-99.

Point 2: Table 1

: add a figure from a good-quality photo of fractures and sections, which are given in table 1

: no differences in typology

: no age difference, combine, not repeat

Response 2: The images in formerly Table 1 have been improved and shown in Figure 3, Page 4.

Point 3: Page 4 Line 113: the order of the figures? fig 3 before fig 2, a mess in the numbering, needs to be corrected

Response 3: The order has been corrected.

Point 4: Page 4 Line 121: in Methodology, we only cite the literature related to the methodology and not the results of research by other authors. This is not acceptable. Delete unnecessary literature. Literature with the results of other authors and related to the results of the article may appear in the Discussion

Response 4: References 27-40 are all our previous research which confirmed that we have often used PIXE in our works. We are the first group in Thailand who use PIXE for materials analysis, especially in archaeological research. However, some literatures have been deleted, Page 17 References 27-33.

Point 5: Page 5 Results: combine the results and interpretation with the discussion into one chapter, the results chapter as it stands does not make sense

Response 5: The parts of Results and Discussion have been combined, Page 6.

Point 6: Page 5 Line 154: in the results we do not provide any method or results quotations, only our own original results and not yet published, because there may be a suspicion of autoplagiarism or plagiarism

: not all test results were described

Response 6: The mentioned content has been moved to the paragraph located between Figure 6 and Table 4. The results described in this paragraph are our own original, Page 12.

Point 7: Page 5 Line 158-160: This is methodology, correct it

Response 7: The content has been moved to the 1st paragraph of Methods, Page 5 Lines 126-128.

Point 8: Page 6 Line 187: add references to geological data

Response 8: The Reference [44] has been added to the content located before Table 3 on Page 9, Line 223.

Point 9: Page 6 Table 2: to the table at the end add the mean values that are shown in the pie charts in fig. 2

Response 9: The average values have been added to Table 2, Page 7.

Point 10: Page 8 Table 3: change the table look like table 2, badly analyze and compare

Response 10:  The format of Table 3 have been revised, Page 9.

Point  11: Page 8 Line 255-259: when writing about the differences, refer to the samples and their images in Fig. 3

Response 11: The samples and images have been referred as the revised content located in the paragraph between Table 3 and Figure 6, Page 9 Lines 236-240.

Point 12: Page 10 Line 271: other samples are also presented

Response 12: The caption of Figure 6 has been revised, Page 11 Lines 256-257.

Point 13: Page 10 Line 278: The results corresponded to SEM. How to correspond, explain it and compare with SEM data.

Response 13: The explanation has been added as the last sentence in the paragraph located between Figure 6 and Table 4, Page 12 Lines 263-265.

Point 14: Page 11-12 Figure 4: What does it mean scale bar? Add information about it. Explain two kind of results: brown and green images

Response 14: The information has been added to the caption of Figure 7, Page 14 Lines 285-286.

Point 15: Page 12 Line 298: The two potteries might: what samples/potteries

Response 15: The content on Page 14 has been revised, Lines 291-292.

Point 16: Page 13 Table 5: explain what does it mean: previous and present works. you need add references

Response 16: The reference [20] has been added to the caption of Table 5, Page 14 Line 297.

Point 17: Page 14 Line 321: these are obvious things

Response 17: This content has been deleted for the conciseness of manuscript.

Point 18: Page 14 Line 330: these are obvious things, methodological things, add summary of these analyses

Response 18: The summary of analyses has been added to the last sentence of the 1st paragraph of Conclusion, Page 15 Lines 310-311.

Point 19: References: needs to be corrected

Response 19:  The references have been corrected.
